

# "Earth-like" planetary magnetotails as non-linear oscillators

Robert J. Burston[1]

[1] Department of Electronic and Electrical Engineering, University of Bath, Bath, BA2 7AY, UK

*Correspondence to*: Robert J. Burston (rjb22@bath.ac.uk)

**Abstract.** A non-linear oscillator model of a simple system analogous to "Earth-like" magnetotail plasmoid formation and release dynamics is presented. In this context, "Earth-like" refers to any magnetosphere with an upstream bow-shock and an elongated downstream tail that undergoes tail plasmoid formation and release. It includes, for the first time in such a model, separate drivers for the Dungey and Vasyliunas Cycles and the capacity to include stochastic and deterministic driving in varying relative and absolute terms. The effects of measurement noise on the model output can also be
simulated. This makes the model suitable to investigate the magnetotail dynamics of Mercury, Earth, Jupiter, Saturn and hypothetical exoplanets with similar magnetospheric configurations. The capacity to predict, in general terms, the behavior of a wide range of stellar-wind – magnetosphere interactions has become even more important in the light of the discovery of thousands of exoplanets in recent years. This model represents the first step towards being able to make such predictions for a wide variety of cases without resorting to detailed modelling of individual cases. It is demonstrated
that the model can exhibit limit cycle (periodic) and chaotic (long-term unpredictable) behavior. The effects of a sufficiently strong dynamical noise component (stochastic driving) are shown to be inherently different from the effects of an equivalent level of simulated observational noise (simulated Gaussian instrument error). The possibilities of chaotic behavior and of dynamical noise dominating the underlying determinism imply that often only short-term forecasting of magnetotail plasmoid formation is possible.

## 1 Introduction

### 1.1 Analogous physical systems

Physics has a long and rich tradition of studying analogous systems, that is, different physical systems that are like each other in one or more significant ways. Such studies can help learning and understanding by relating unfamiliar physical situations to ones that are familiar through previous study or everyday experience. A very fruitful avenue has been the use of superfluids as analogies to cosmological and quantum mechanical 25 systems. The fundamental value of superfluid analogies is that it is in general much easier to conduct experiments on superfluids than on the systems the analogies are made with. Just some examples are given in the much cited review, (Volovik, 2001). Two more examples date further back in



the history of physics. These are the analogies first, between classical electromagnetism and Newtonian mechanics and, second, between classical electromagnetism and fluid dynamics. Both were first presented by James Clerk Maxwell in his two volume Treatise on Electricity and Magnetism (Maxwell, 1873). The electrical – mechanical analogy has proved extremely useful on a practical level for designing band-pass filters. See (Mason, 1922) for an elementary introduction to the topic. The space physics community, therefore, should embrace the use of analogous systems in its pursuit of new understandings. In some respects it already has: ionospheric and magnetospheric analogies to electrical circuits are standard textbook fare e.g. (Hargreaves, 1992;R.D. Hunsucker and Hargreaves, 2003;Robert W. Schunk and Nagy, 2000). More specifically and most pertinent to this study, tail reconnection and plasmoid formation in Earth's magnetosphere have been treated as analogous both to an electrical circuit and to the release of water drops from a leaky tap, but before discussing applications of these analogies a brief description of their context is given.

**1.2 Planetary magnetospheres, tail reconnection, plasmoid formation and the Dungey and Vasyliunas Cycles**

Planetary magnetospheres can be divided into two classes, intrinsic and induced. If a planet has an internally generated magnetic field, then this field's interaction with the stellar wind and Interplanetary Magnetic Field (IMF) flowing past it produces an *intrinsic* planetary magnetosphere. An *induced* planetary magnetosphere is created when the surface or atmosphere of a planet that has no internally generated magnetic field interacts with a stellar wind and IMF flowing past it. In the solar system, the planets Venus and Mars have *induced* magnetospheres. Mercury, Earth, Jupiter, Saturn, Uranus and Neptune all have appreciable *intrinsic* magnetospheres. Of the latter group, the first four can all be described as having a similar over-all magnetospheric configuration consisting of a compressed magnetic field on the day side and a long stretched-out "magnetotail" on the night side. This configuration is henceforth described as "Earth-like" for convenience. Other configurations of intrinsic planetary magnetosphere are possible, as demonstrated by Uranus and Neptune (Keiling, 2015).

Earth's magnetotail can, under conditions of southward IMF, undergo a cyclical process in which energy from the solar wind is stored in the magnetotail by way of stretching and compression of the magnetic field, followed by the release of a plasmoid which travels downstream with the solar wind flow. The magnetotail subsequently returns to a more nearly dipole field configuration, before undergoing further stretching and compression and repeating the cycle. This process is driven by the Dungey Cycle (Dungey, 1961). One cycle of the process as a whole generates a substorm at Earth (Akasofu, 1964).

There is evidence that processes of magnetotail stretching, plasmoid formation and detachment and subsequent relaxation to a more nearly dipole configuration also occur at Mercury, Jupiter and Saturn. This means that all the Earth-like magnetospheres in the solar system have been observed to do this (Keiling, 2015). However, in addition to the Dungey Cycle there is another mechanism that can drive plasmoid generation in Earth-like



magnetotails, known as the Vasyliunas Cycle. The Dungey Cycle is powered by the solar wind. Contrastingly, the Vasyliunas Cycle is driven by the co-rotation of plasma in the atmosphere of a planet with an intrinsic magnetosphere. The energy source ultimately derives from the kinetic energy of rotation of the planet's atmosphere (Vasyliunas, 1983).

The absolute and relative energy input from each Cycle varies from planet to planet. At Mercury and Earth, the Vasyliunas Cycle is negligible and the Dungey Cycle is dominant. At Jupiter the Vasyliunas Cycle dominates. Saturn is observed to have significant contributions from both Cycles (Keiling, 2015). In order to obtain a general understanding of Earth-like magnetotail dynamics, therefore, one must take account of both Cycles and their relative contributions to the dynamics in each case.

This is also crucially important if one looks further afield, to extra-solar planets (exoplanets), which have been discovered by the thousands, orbiting various star types, with a large range of masses and orbital parameters, e.g. (Hatzes, 2016;Winn and Fabrycky, 2015). The possible range of stellar wind/IMF conditions, planetary magnetic field strengths, orbital distances, planetary radii and rotational periods that might plausibly combine to form Earth-like magnetospheres is therefore vast. The consequent range of possible relative and absolute strengths of the Dungey and Vasyliunas Cycles is also vast.

Faced with the problem of developing a general understanding of the possible range of magnetotail dynamics, even when restricted to Earth-like cases, a simple model that can be easily adapted to represent numerous different classes of planet and stellar wind/IMF conditions has the advantage over more complex and computationally demanding methods, such as magnetohydrodynamic (MHD) or kinetic theory plasma models. Such models generally take of the order of years to develop and test, have been designed for one specific case and are not easily adaptable to radically different cases. In attempting to develop a model applicable to such a wide range of cases, detail (the strength of MHD and kinetic models) must inevitably be sacrificed in order to increase flexibility and speed. This is where analogous systems can help.

### 1.3 Physical analogies to magnetotail plasmoid formation and the substorm current wedge

#### 1.3.1 The leaky tap analogy

The analogy between plasmoid formation and the formation and release of a water drop by a leaky tap was first put forward by (Hones, 1979) purely as an aid to conceptualising and visualising the process and there it rested until the publication of (Baker et al., 1990). In the intervening time important developments were being made in the field of non-linear dynamics, both experimental and theoretical, leading to widespread



awareness and application of chaos theory. Crucial for this context was the demonstration of the potential for chaos in the time series of intervals between drops released by a leaky tap in (Shaw, 1984). In that short monograph Shaw makes his own analogy ("the Shaw Model", henceforth) between the dripping tap and a steadily increasing mass on a spring that, when a certain condition is met, loses some mass (equivalent of a water drop). He showed that, depending on the rate of increase of mass (equivalent to the rate of water leakage from the tap) the time series of drip release

85    times could show regular periodicity or deterministic chaos. Hence, he was able to reduce a problem that from first principles requires solution of the Navier-Stokes Equation, with infinite degrees of freedom, to one with only three degrees of freedom. The reason this is possible is because the vastly dominant degrees of freedom are contributed by the non-equilibrium interaction between the gravity, the water's surface tension and the water's mass. The infinite number of microscopic degrees of freedom can be entirely neglected. The previously mentioned (Baker et al., 1990) paper then argued, if magnetotail dynamics is like leaky tap dynamics and leaky tap dynamics is like non-linear spring oscillator dynamics then

magnetotail dynamics should be like non-linear spring oscillator dynamics. They proceeded to describe a model ("the Baker Spring Model") of magnetotail dynamics based on this principle. The Baker Spring model does not closely resemble the Shaw Model except in so far as both are what are termed "relaxation oscillator" models, which all have the characteristic of a slow build up to a sudden change of state of the system. In this three-way analogy there is a slow build-up of magnetic potential energy / suspended water / mass on the spring – followed by a sudden release of a plasmoid / a drop of water / a discrete amount of mass. Then, the magnetic field becomes rapidly less stretched / the water surface tension is

reduced / the spring tension is reduced i.e., in all three cases the system "relaxes". This sequence then repeats in a cyclical manner. The dripping tap analogy is therefore a useful one for assessing qualitatively the range of possible dynamical modes of behaviour of Earth's magnetotail. (Later, (Freeman and Morley, 2004) created a simple oscillator model and assumed that substorm repeat times are constant for constant solar wind conditions. All variability is assumed to derive from variations of the solar wind parameters.)

### 1.3.2 The substorm current wedge electrical circuit analogy

As already mentioned, the idea of a set of magnetospheric-ionospheric current systems is established textbook physics. These are all an analogy for the motion of charged particles in Earth's atmosphere and near-Earth space, because the concept of electrical current is itself always just an analogy between fluid flow and the motion of charged particles. In response to the Baker Spring Model, (Klimas et al., 1992) a new model of the substorm current wedge, the Faraday Loop Model (Klimas et al., 1992), was constructed which consisted, mathematically, of a "non-linear, damped harmonic oscillator."

Subsequent work used various forms of substorm current wedge based approaches. (Pavlos et al., 1994) developed another model based on principles from MHD and then (Horton and Doxas, 1996) used energy conservation principles in the magnetotail for a model, expanding from that



to include the entire substorm current wedge circuit in (Horton and Doxas, 1998). Next came the solar-wind – magnetosphere – ionosphere model known as WINDMI, which derived from a kinetic theory approach to the substorm current wedge circuit but maintains low dimensionality compared to a full kinetic model (Horton et al., 1999). This model has expanded in sophistication over subsequent years (Andriyas et al., 2012;Horton et al., 1999;Smith et al., 2000;Spencer and Patra, 2013).

The motivation for all the substorm current wedge models is to take a step beyond the qualitative dynamics of the leaky tap analogy to something more quantitative.

### 1.3.3 Analogous models are intentional simpler than the first-principles problem

Common to all these approaches, whether based on the leaky tap or electrical circuit analogy, is the fact that they are enormously simpler than trying to model the system from first principles (whether MHD or plasma kinetic theory): they offer the properties of flexibility and lower computational demands suggested as essential for a general model of Earth-like magnetotails in section 1.2 above. They also all have one drawback in terms of use for the general case: because they were developed specifically with Earth in mind, none takes any account of contributions to the dynamics by a Vasyliunas Cycle. Hence, unmodified, none is useful for the general case. This paper presents a model that returns to the leaky tap analogy but for the first time features separate inputs equivalent to a Dungey Cycle and a Vasyliunas Cycle, that can be independently adjusted in terms of absolute and relative contributions, making it the first general model of the qualitative dynamics of Earth-like magnetotails. But what is "qualitative" dynamics?

### 1.4 Different types of dynamics

Systems can be broadly classified as stochastic or deterministic i.e. either behaving randomly, with no greater forecasting possibility than can be derived from the underlying probability distribution, or evolving with respect to time according to an equation or a set of equations that may or may not be known. This is a fundamental qualitative difference. There are many varieties of subcategories and combinations of these and the relevant ones in the present context are discussed below. See e.g. (Strogatz, 1994) for a good introduction to dynamical systems theory in general, other qualitative types of dynamics and more detail and examples of the dynamical types discussed below.

### 1.4.1 Limit cycles

Limit cycles are closed loops in the phase space of a dynamical system. For example, for any initial conditions, a spring and mass obeying the ideal (no damping) Hooke's Law will describe an ellipse in velocity-displacement phase space that it will repeat forever. This ellipse is a simple example





of a limit cycle. All limit cycles represent periodic behaviour in their parent systems. However complicated the trajectory through the system's phase space might be, if it ever returns to exactly the same point as one it has been to before, then a closed loop is formed and the system has a limit cycle for that particular set of control parameters and initial conditions. In the ideal Hooke's Law case, the system starts on a limit cycle, regardless of the values of the control parameter and initial conditions. For dynamical systems in general, this is not the case. In other systems it may be that, though a limit cycle exists, the chosen set of initial conditions is not on it. Over time, however, the system's trajectory may take it to the limit cycle where it then remains forever. This is known as an attracting limit cycle, as it attracts to itself the trajectories in phase space surrounding it. Limit cycles are not the only kind of attracting structure that can occur in the phase spaces of dynamical systems. In general, such structures are termed "attractors." The region of phase space in which all initial conditions lead to an attractor is called the "basin of attraction." Basins of attraction may be finite or infinite in volume depending on the system. The part of a trajectory through phase space that comes before arrival on an attractor of any kind is called a "transient."

### 1.4.2 Deterministic chaos

Deterministic chaos (hence forth, simply "chaos") is a term reserved for a particular type of behaviour that superficially appears stochastic but is in fact purely deterministic. Many systems can display this type of behaviour given the correct circumstances. Some of these systems are very simple but all of them have a non-linear component. Every case of chaotic dynamics will display a particular kind of structure in phase space called a "strange attractor." As the name implies these structures are both attractors and very strange indeed. They bear a superficial resemblance to limit cycles but when viewed in close up it is seen that they are not closed loops. No matter where one arrives on a strange attractor, the trajectory never crosses itself, even after infinite time, but it also never leaves a finite volume of phase space, either. If a trajectory in phase space is on a strange attractor, the system itself will exhibit apparently random behaviour, but within certain limits that may not be the absolute physical limits of the system.

### 1.4.3 Instrument noise

Instrument noise is simply the random component of an observation introduced due to finite instrument precision. It will always be present in any set of scientific observations, regardless of whether the observed system is deterministic or stochastic. It has the effect of adding a stochastic component to trajectories in phase space. Depending on the exact probability distribution, instrument noise can range from negligible to so dominant that any deterministic component of the dynamics is completely masked. In phase space, this manifests at low levels as a blurring of the intrinsic





dynamical structures. As the noise level increases, so the blurring worsens until any underlying deterministic dynamics becomes undetectable. Whilst not strictly a type of dynamics, it can be important because of the uncertainty it introduces about the system's real behaviour.

**1.4.4 Dynamical noise**

Imagine a system is governed by a set of deterministic equations but the control parameter(s) may change with respect to time. The result is a system experiencing dynamical noise. In the Shaw Model this would be exemplified by changing the rate of mass increase randomly. In the leaky tap analogy this would be random alterations in the water flow rate and in the magnetotail case it would be random changes in the rate of accumulation of magnetic flux in the magnetotail. (See e.g. (Kantz, 1997) for more detail on dynamical and instrument noise in the context of data

analysis.) Dynamical noise can complicate the understanding of dynamical systems as it may cause different types of phase-space structures to be present in different segments of a single series of observations of a dynamical variable. For example, in the case of the leaky tap, experiment and theory both show that switching between limit cycles with different periodicities is possible, as is switching from limit cycle behaviour to chaos, just by changing the water flow rate.

**1.5 Types of dynamics that might be occurring during plasmoid formation and release**

**1.5.1 Limit cycles**

If the timing of plasmoid releases from an Earth-like magnetotail shows periodicity, then the phase space of the system would contain a limit cycle. Note that the period need not be a single-step one. A series with arbitrary time units, $\tau$, having events occurring according to the pattern, $3\,\tau$, $2\,\tau$, $1\,\tau$, $3\,\tau$, $2\,\tau$, $1\,\tau$ … is as much a periodic one as a series with events at $2\,\tau$, $2\,\tau$, $2\,\tau$... The number of steps before the sequence repeats can be arbitrarily long but not infinitely long. If a periodicity is observed in the intervals between plasmoid releases, then a limit cycle in phase space is

175 implied and the dynamics of the system are definitively deterministic.

**1.5.2 Chaos**

A time series of observations that shows no exact periodicity might be governed either by a stochastic process or a chaotic one. Telling the difference by simply examining the series can be difficult but in phase space the difference becomes clear – either the entire phase space is filled (stochastic) or a strange attractor is seen (chaotic). Examination of proxies for plasmoid releases in Earth's magnetotail show periods of apparent randomness,

implying stochastic or chaotic dynamics may be occurring. Other segments of the time series of the same proxies show evidence of periodicity, implying limit cycle dynamics. There are also fallow periods when plasmoid formation is suppressed (Baker et al., 1990). The existence of periodic, a-periodic and absent plasmoid formation at different times indicates changes in the driving of the system – dynamical noise. Since instrument





noise is omnipresent, all the qualitative types of dynamics described in section 1.4 could plausibly be operating in Earth's magnetotail and hypothetically in that of other planets and exoplanets with Earth-like magnetospheres.

**1.6 A non-linear oscillator analogy model of general Earth-like magnetotail dynamics**

This paper presents a new non-linear oscillator model of Earth-like magnetotail plasmoid release dynamics. It includes a Vasyliunas Cycle as well as a Dungey Cycle and also incorporates dynamical and instrument noise effects. No previous model includes all of these. The (Freeman and Morley, 2004) model has considered dynamical noise previously but no others do. None has a Vasyliunas Cycle. None considers instrument noise.

This new model is an extension of the Shaw Model. The new model meets the criteria set out in section 1.2 and, whilst it cannot offer quantitative results, the strong analogy with plasmoid formation and release allows for examination of the range of possible qualitative dynamics that can be expected in Earth-like magnetotails. Full details of the model are given in section 2.

**1.6.1 Simulations**

Using this new model in various simulations reveals attracting limit cycles and strange attractors exist in the system's phase space, depending on the precise control parameter settings. Additionally, the model can be switched from limit cycle to chaotic behaviour solely by adjusting the rate of mass loading (equivalent to changing the total contribution of the Dungey and Vasyliunas Cycles). This matches the types of behaviour observed in Earth's magnetotail. Dynamical noise is demonstrated to manifest significantly differently from simulated instrument noise of equivalent proportional scale.

**1.7 Implications**

The intrinsic non-linearity of Earth's solar-wind – magnetosphere – ionosphere coupling has led to a considerable body of work investigating the possibility that the dynamics involved is globally a low-dimensional chaotic system – a Chaotic Magnetosphere Hypothesis (CMH) e.g. (Athanasiu et al., 2003;Balikhin et al., 2001;Consolini et al., 1996;Horton, 1997;Horton et al., 2001;Sharma et al., 1993;Voros, 1994;Zivkovic and Rypdal, 2011;Pavlos et al., 1999). A counter proposal is that the dynamics are stochastic, specifically displaying a bi-coloured noise spectrum – a Noisy Magnetosphere Hypothesis (NMH) (Takalo et al., 1993, 1994). The two proposals are not, in fact, mutually exclusive: the system may have a dynamical noise component overlying a deterministic component and the deterministic component may behave chaotically under the right conditions – a Noisy Chaotic Magnetosphere Hypothesis (NCMH). This hypothesis is put forward for the first time here and has not been studied



before. Even if there is no dynamical noise, any attempt to demonstrate that the system is chaotic from observations must take into account instrument noise. Hence noise cannot be ignored even if the system is chaotic and a pure CMH must be abandoned in favour of the NCMH.

The question of the NMH verses NCMH is of practical importance, because magnetotail plasmoid formation has several dramatic consequences on Earth. Charged particles are accelerated along magnetic field lines into the polar ionosphere, where they trigger aurorae, cause fluctuating magnetic fields at the Earth's surface, heat the thermosphere and cause build-up of electrical charge on satellites in Low Earth Orbit (LEO). Such static charge build-ups are potentially damaging to satellites and their on-board instrumentation (Fennell et al., 2001). Thermospheric heating increases the drag on LEO satellites, which then have to use propellant from their finite on-board stores in order to maintain their correct orbital altitude, thus reducing

their operational lifetimes (Qian and Solomon, 2012). On the ground, the magnetic fluctuations can cause Geo-magnetically Induced Currents (GICs) in long metallic structures such as pipelines and electrical power lines. Such current surges are potentially hazardous, posing risk of fire and transformer failure if they are large enough (Gaunt, 2016). Auroral phenomena are associated with degradation of performance of Global Navigation Satellite systems such as the Global Positioning System, which can have knock-on effects on users of such systems (Kinrade et al., 2012). If the system driving these phenomena is either chaotic or highly stochastic, forecasting these hazards is intrinsically limited to the short-term only.


In the case of exoplanets, where detection of auroral emissions may become possible when the Square Kilometre Array radio telescope becomes fully operational, these factors will be crucial in the analysis of received signals (Nichols and Milan, 2016).

## 2 Method

### 2.1 The model definition

The model presented here is an extended and modified version of the Shaw Model. The magnetic field of the magnetotail is represented by a mechanical spring which is stretching and oscillating as mass is continuously added to it. When the spring reaches a critical displacement, an amount of mass proportional to the instantaneous velocity is lost. This lost mass represents the detachment of a plasmoid. Mathematically, the model is defined by the set of equations:

$$F = m\frac{d^2x}{dt^2} = mg - kx - C_{damp}\frac{dx}{dt} - C_{drag}\left(\frac{dx}{dt}\right)^2, \qquad (1)$$

$$\frac{dm}{dt} = C_D \pm w_D + C_V \pm w_V, \qquad (2)$$






$$\Delta m = \left. C_u \frac{dx}{dt} \right|_{x=x_c},$$
( 3 )

Equation ( 1) represents the forces on the spring (from left to right, gravity, the elastic restoring force, damping by heat generation and viscous drag). $F$ is the total force, $m$ is the attached mass (not constant), $k$ is the spring constant, $x$ is the displacement, $t$ is time and $C_{damp}$ and $C_{drag}$ are the damping and drag constants, respectively. Note that all parameters usually associated with a viscous drag term have been subsumed into the

one constant; the details are not relevant in this context. Eq. ( 2) describes the continuous rate of change of mass and has four terms. $C_D$ and $C_V$ are the rates of increase of attached mass due to the Dungey and Vasyliunas Cycles, respectively. The time dependence of the values of $C_D$ and $C_V$ may be altered arbitrarily in the model. In this paper, however, each is set to be constant for the whole of any given model run. $w_D$ and $w_V$ are noise terms that modify the values of $C_D$ and $C_V$. These may be set to zero (purely deterministic case) or to select from any pseudo-random distribution realisable in Matlab at each time step of the model run. The distributions or even the types of distribution need not be the same, in general, i.e.

$w_D \neq w_V$ if desired. These variations represent dynamical noise. Equation ( 3) represents the instantaneous loss of mass when a water drop/plasmoid detaches at the critical displacement, $x_c$. $\Delta m$ is the instantaneous mass lost. $C_u$ is a constant of proportionality that may take any positive finite value but is fixed for any given model run. This model can be reduced to the Shaw Model by setting $C_{damp}$, $C_{drag}$, $C_V$, $w_D$ and $w_V$ to equal zero. The model is a completely deterministic initial value problem if $w_D$ and $w_V$ are set to zero but is intrinsically non-linear because of Eq. ( 3). The model is implemented in MATLAB code using the "ode45" numerical differential equation solver. This solver is in turn an implementation of a

Runge-Kutta numerical integration scheme (Dormand and Prince, 1980;Shampine and Reichelt, 1997). Every time the critical extension, $x_c$, is exceeded, solution of Eq. (1) and Eq. ( 2) is interrupted, the retained mass is recalculated according to Eq. ( 3) and the model run is then resumed. The new, reduced mass and the previously pertaining velocity and displacement values are used as a new set of initial values for the ode45 solver. If at any time $\Delta m > m$, a non-physical negative value of $m$ is avoided by setting $\Delta m = m$. Measurement noise is easily simulated by taking the output sequences of mass, mass lost, velocity and displacement and adding a normally distributed error value to each individual number in the

series. In this paper this is only done for specific cases of the $\Delta m$ time sequence.

**2.2 Types of dynamical behaviour**

An unsystematic but wide-ranging exploration of the *parameter* space of the model was made. Two classes of general behaviour were identified and a specific example of each is presented in section 3. These behaviours are i. approach to limit cycles (model parameters for an example are given in Table 1) and ii. deterministic chaos (model parameters for an example are given in Table 2).






To be chaotic a system must be both fully deterministic and highly sensitive to its initial conditions. The model given by Eqs. ( **1**), ( **2**) and ( **3**) is completely deterministic if $w_D$ and $w_V$ are set to equal zero. If $w_D$ or $w_V > 0$, then the system can be considered a deterministic one with additional dynamical noise. In order to demonstrate conclusively that the system can be chaotic an example with $w_D = w_V = 0$ is shown.

A chaotic system is characterised by "sensitive dependence on initial conditions". More rigorously, initially arbitrarily close trajectories in phase space must on average diverge exponentially from each other in at least one dimension. Note that this divergence should be measured when the trajectories are on the attractor, not during any initial transient period and that the exponential divergence cannot be sustained indefinitely because of the finite size of the attractor (Strogatz, 1994). In order to test for this condition the following method was adopted: The model parameter values were held constant across three runs. These conditions were variations away from the initial conditions given in Table 2. The variations are random

numbers taken from normal distributions with standard deviations, $\sigma = 0.1\%$ of the values given in Table 3.

| Para-meter | $g$ ($ms^{-2}$) | $k$ ($kgs^{-2}$) | $C_{damp}$ ($kgs^{-1}$) | $C_{drag}$ ($kgm^{-1}$) | $C_D$ ($kgs^{-1}$) | $C_V$ ($kgs^{-1}$) | $w_D$ ($kgs^{-1}$) | $w_V$ ($kgs^{-1}$) | $C_u$ ($kgs^{-1}$) | $x_c$ (m) |
|---|---|---|---|---|---|---|---|---|---|---|
| Value | 10 | 1 | 0 | 0 | 0.007 | 0.007 | 0 | 0 | 10 | 5 |

**Table 1: The model parameters for the run illustrated in Figs. 1–3. (Approaching a limit cycle.)**

| State Variable Initial Conditions | $x_0$ (m) | $v_0$ ($ms^{-1}$) | $m_0$ (kg) |
|---|---|---|---|
| Value | 1 | 10 | 0.4 |

**Table 2: The initial conditions of the model for the run illustrated in Figs. 1–3. (Approaching a limit cycle.)**

| Para-meter | $g$ ($ms^{-2}$) | $k$ ($kgs^{-2}$) | $C_{damp}$ ($kgs^{-1}$) | $C_{drag}$ ($kgm^{-1}$) | $C_D$ ($kgs^{-1}$) | $C_V$ ($kgs^{-1}$) | $w_D$ ($kgs^{-1}$) | $w_V$ ($kgs^{-1}$) | $C_u$ ($kgs^{-1}$) | $x_c$ (m) |
|---|---|---|---|---|---|---|---|---|---|---|
| Value | 10 | 3.08 | 0 | 0 | 0.003 | 0.003 | 0 | 0 | 0.12 | 3 |

**Table 3: The model parameters for the run illustrated in Fig 4. (Deterministic chaos.)**



| State Variable Initial Conditions | $x_0$ (m) | $v_0$ (ms$^{-1}$) | $m_0$ (kg) |
|---|---|---|---|
| Value | 1 | 1 | 0.01 |

**Table 4: The initial conditions of the model for the run illustrated in Fig 4. (Deterministic chaos.)**


The evolution of the separation of the trajectories over time was calculated in the following way: The Euclidean distance in the system's phase space is defined as $\Delta S = \sqrt{\Delta m^2 + \Delta v^2 + \Delta x^2}$ and the three trajectories are labelled A, B and C. The separations between trajectories at any given time can be calculated from the vectors, $\boldsymbol{r}_A - \boldsymbol{r}_B$, $\boldsymbol{r}_A - \boldsymbol{r}_C$ and $\boldsymbol{r}_B - \boldsymbol{r}_C$, where $\boldsymbol{r}_A, \boldsymbol{r}_B, \boldsymbol{r}_C$ are the position vectors in phase space of the trajectories, A, B and C. These separations are then plotted and an exponential curve is fitted to their outer envelopes.

## 2.2 The effect of increasing the total loading rate, $\frac{dm}{dt}$, (Eq. ( 2 ) ).

Changing the mass loading rate, which is analogous to changing loading by the Dungey and Vasyliunas Cycles, whilst keeping all other parameters constant, may affect the mode of dynamical behaviour that the system falls into and is therefore investigated by the following simulation experiment: For each run, $w_D = w_V = 0$, so that there is no dynamical noise. Two runs were made, each with $C_D = C_V > 0$. Run 1 had $C_D = C_V = 2.5 \times 10^{-5}$. Run 2 had $C_D = C_V = 2.5 \times 10^{-4}$, one order of magnitude larger. All other parameters were kept constant across both runs. The model parameters

used for this experiment are set out in full in Table 3.

| Para-meter | $g$ (ms$^{-2}$) | $k$ (kgs$^{-2}$) | $C_{damp}$ (kgs$^{-1}$) | $C_{drag}$ (kgm$^{-1}$) | $C_D$ (kgs$^{-1}$) | $C_V$ (kgs$^{-1}$) | $w_D$ (kgs$^{-1}$) | $w_V$ (kgs$^{-1}$) | $C_u$ (kgs$^{-1}$) | $x_c$ (m) |
|---|---|---|---|---|---|---|---|---|---|---|
| Value (Run 1) | 10 | 1 | 0 | 0 | 2.5x10$^{-5}$ | 2.5x10$^{-5}$ | 0 | 0 | 10 | 5 |
| Value (Run 2) | 10 | 1 | 0 | 0 | 2.5x10$^{-4}$ | 2.5x10$^{-4}$ | 0 | 0 | 10 | 5 |

**Table 5: The model parameters for the two runs illustrated in Fig. 9. The limit cycle case is given by the top pair of values of $C_D$ and $C_V$. The chaotic case is given by the bottom pair of values of $C_D$ and $C_V$. All other parameters and the initial conditions were the same for both model runs.**






| State Variable Initial Conditions | $x_0$ (m) | $v_0$ (ms$^{-1}$) | $m_0$ (kg) |
|---|---|---|---|
| Value (Run 1) | 1 | 10 | 0.4 |
| Value (Run 2) | 1 | 10 | 0.4 |

**Table 6: The initial conditions for the two model runs illustrated in Fig. 9. The limit cycle case is given by the top pair of values of $C_D$ and $C_V$. The chaotic case is given by the bottom pair of values of $C_D$ and $C_V$. All other parameters and the initial conditions were the same for both model runs.**

**2.3 A comparison between the effects of increasing levels of dynamical noise and equivalent levels of measurement noise on the lost mass time series.**

In all the previous experiments the dynamical noise components of the loading rate ($w_D$ and $w_V$ in Eq. ( **2**) ) have been set equal to zero. The effect of setting these to be random numbers selected from a Gaussian distribution of zero mean and standard deviation, $\sigma$, was investigated. This was compared with the effects of simulated instrument noise. For the instrument noise case, a random number taken from a Gaussian distribution of zero mean and standard deviation, $\sigma$, was added to each value of $\Delta m$, using the results from the dynamical noise free case specified by Table 3. The values of $\sigma$ used were scaled according to the value of the mass lost, $\Delta m$, such that $\sigma = O(\Delta m) - 2$, for one run and $\sigma = O(\Delta m) - 1$ for

another, where $O(\ )$ implies "order of magnitude of". This scaling matches that of the dynamical noise case, where the noisy runs had $\sigma = O(C_D + C_V) - 2$ and $\sigma = O(C_D + C_V) - 1$, respectively. The proportionate levels of noise added are therefore the same for the dynamical and instrument noise cases and the results are directly comparable. The values of the other parameters used for these runs are the same as those in Table 3.

**3 Results**

Readers are strongly encouraged to view the larger versions of the figures in section 3 that are held in the "supplemental material" for this paper. This is especially useful for understanding the phase space diagrams. Supplemental figure numbers correspond to the figure numbers in the main text.



### 3.1 Types of dynamical behaviour

The possible types of dynamical behavior are best examined in terms of the trajectories of the system through its *phase* space. For this model the
phase space is 3-dimensional, with the state variables, displacement, velocity and mass as the axes. An unsystematic but extensive search of the model *parameter* space found the following types of dynamical behavior: i. Approach to limit cycles. ii. Strange attractors.

### 3.1.1 Attracting limit cycles

The full model given by Eqs. ( **1**), ( **2**) and ( **3**) can exhibit attracting limit cycle behaviour given certain parameters and initial conditions. An example of a simulation showing behaviour consistent with an attracting limit cycle is given by the parameters and settings shown in tables 1 and
2. The phase space trajectory of the model run is illustrated in Fig. 1. This is a scatter plot, not a line plot, with successive points 1/10th of a second apart. The passage of time is indicated through the use of the colour bar. Earlier times are indicated by darker tones. It is perhaps easier to see that the trajectory approaches a limit cycle from the plots verses time of the individual state variables, velocity, displacement and mass (Fig. 2). The outer envelopes of the state variables all tend to constants with respect to time (i.e. tend to horizontal lines) as the variables approach the limit cycle. The rapid convergence to a repeating cycle can be seen very clearly in the graph of the instantaneous mass *lost* ($\Delta m$) verses time (Fig. 3). Note that
the time axis is plotted on a logarithmic scale. The mass is lost at discrete intervals marked with crosses. The black line linking the crosses is a visual aid to help convey how rapid the convergence is. This line has become horizontal in less than $10^3$ seconds and if one could only measure this aspect of the system it would appear to have reached the limit cycle in this time. This is faster than the envelope lines of Fig. 2 become horizontal, indicating that this is a less sensitive measure of the dynamics than the state variables themselves. The mass lost has become almost indistinguishable from its limiting value in only three steps.

### 3.1.3 Deterministic chaos


The other type of behaviour observed with this model is that of deterministic chaos. The phase space trajectory of the model run using the parameters and initial conditions given in Tables 3 and 4 is illustrated in Fig. 4. The distribution of points in the phase space with respect to time is markedly different from that shown in Fig. 1. Instead of a progression towards a limit cycle, the trajectory wanders over the entire region containing the attractor. The approximately helical structures visible in the attractor (and in the limit cycle of Fig. 1) can be explained by considering the governing
equations. For the ideal Hooke's Law, the phase space trajectory must be an ellipse, as already mentioned. When the variation in mass given by Eq. ( **2**) is added to the system, the ellipse will be stretched in the third dimension and an approximately helical structure will form. When the instantaneous loss of mass given by Eq. ( **3**) occurs the trajectory drops back to near the zero-mass plane and the process repeats. In the limit cycle case, each repetition brings the trajectory closer to a condition which will repeat exactly forever. In the case of a strange attractor there is never an




exact repetition. Figure 4 has this character. The system jumps from one mode to another with a different base radius when the discontinuous

changes in mass occur. Each mode has a different helix-like structure associated with it.

Figure 5 shows the mass lost with respect to time for the case given by Tables 3 and 4. From panel B it can be seen that the system arrives on the attractor at between 200 – 300 s, the same order of magnitude as for arrival on the limit cycle in the previous case. (Compare with Fig. 3). There is clear structure in the graph. Rather than an apparently random distribution of amounts of mass lost, most cases are one of only four different discrete

values of lost mass, each corresponding to the mass at the top of one of the approximate helices seen in Fig. 4. However, there are exceptions and not only during the initial transient period, e.g. at approximately $t = 3.2 \times 10^4$ s, m = 0.75 kg (most clearly seen in panel A). Panel A also most clearly shows that the system is not just repeating a long-period cycle. The time intervals between mass losses of approx. 2.75 kg appear random.

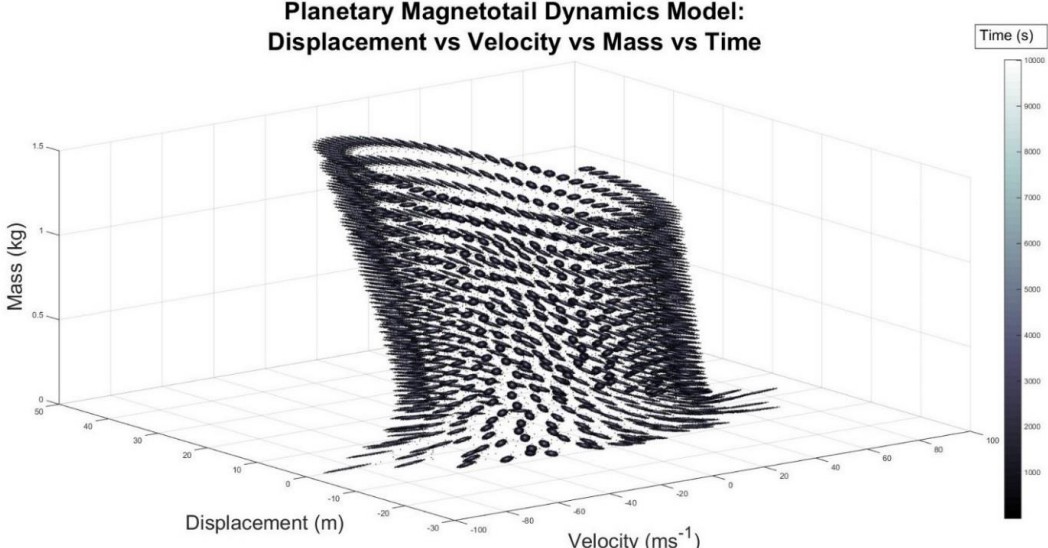

**Figure 1: The trajectory in phase space of a model run approaching a limit cycle. Over time the trajectory in phase space approaches and settles on an exactly repeating loop indicated by the lightest points in the figure, seen at the center of each disc-like structure making up the larger over-all distorted helix. See supplemental figure 1 for a zoom-in, showing the detailed structure of this attractor.**

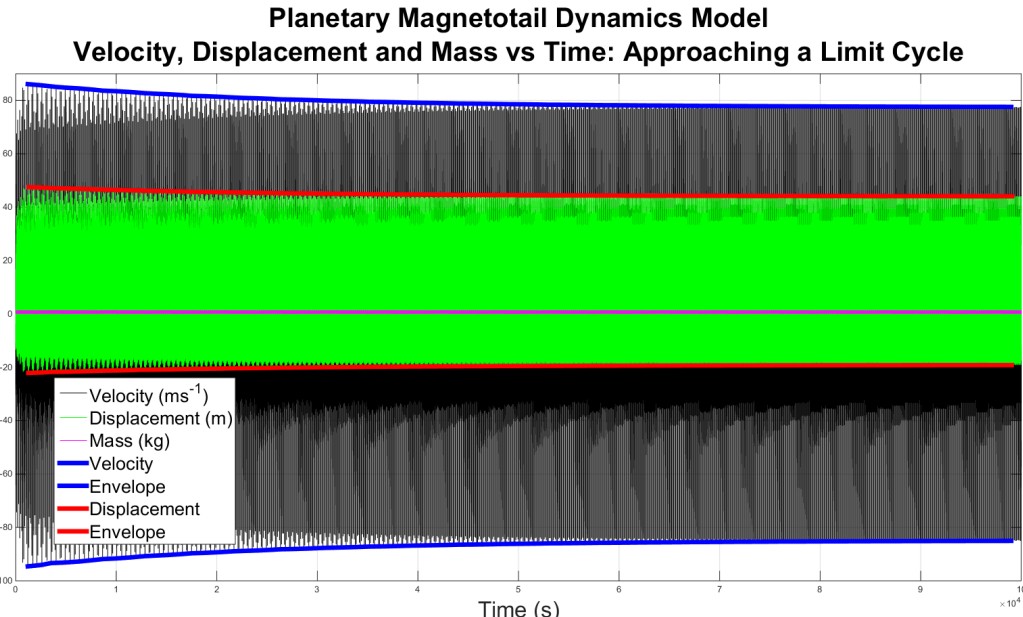

**Figure 2: The state variables, velocity, displacement and mass verses time as the system approaches a limit cycle. The outer envelope of the velocity variation is shown as the thick blue lines. The outer envelope of the displacement variation is shown as the thick red lines. When these become horizontal the system has reached its limit cycle. The variation in retained mass is much smaller and not visible on this scale but follows a similar pattern to that of the velocity and displacement.**





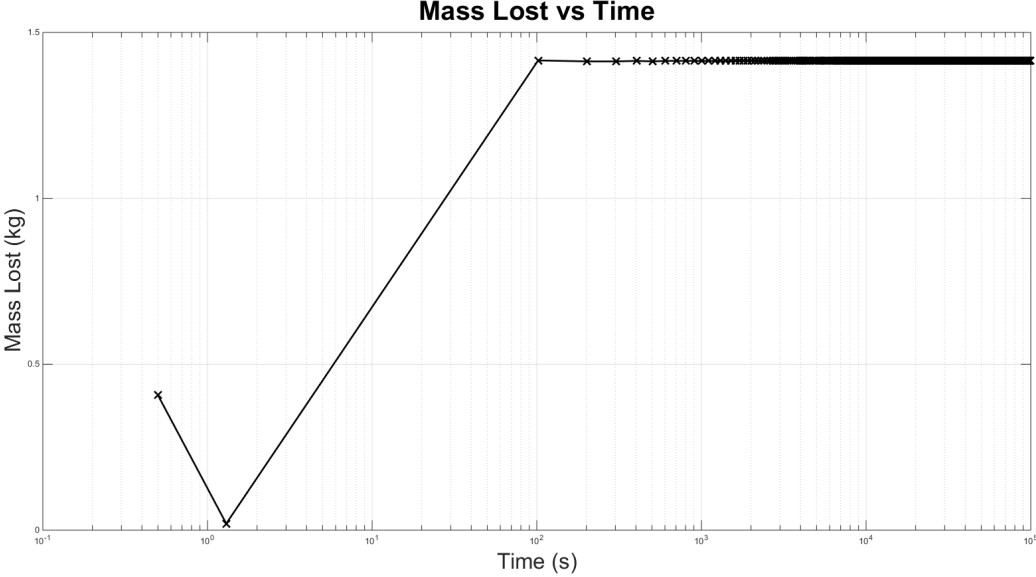

**Figure 3: Mass lost verses time plot for a limit cycle. (Note the logarithmic scale of the time axis.) The mass is lost at discrete times marked by crosses. The line linking the crosses is a visual aid. When the line becomes horizontal ($< 10^3$ s) the system appears to have attained its limit cycle. Note that this happens much more quickly than for the state variables, velocity, displacement and (retained) mass (see Figure 2). This implies that the mass lost is a less sensitive indicator of the dynamics than the state variables are.**


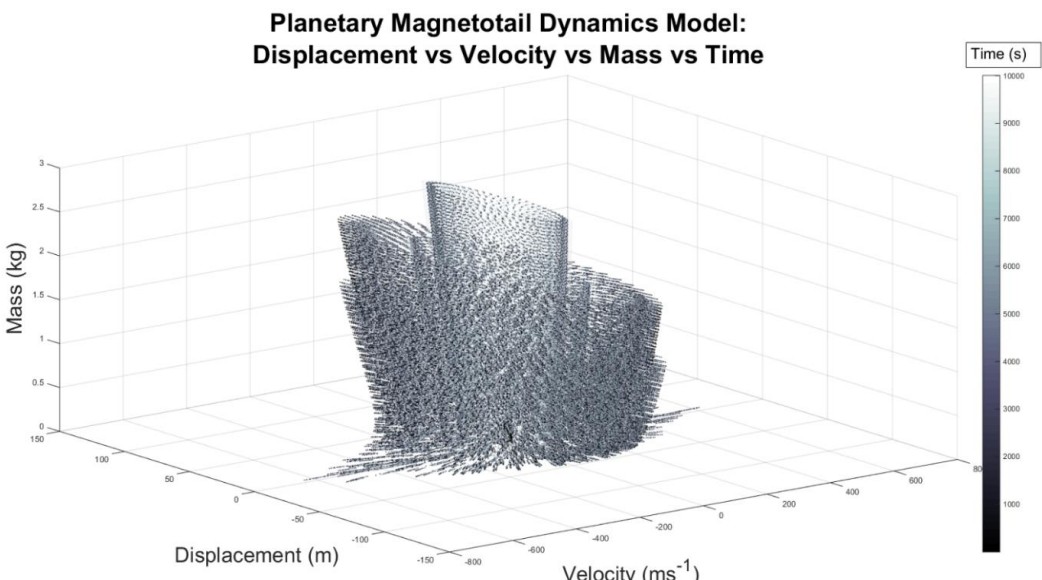

**Figure 4: A strange attractor in the model phase space. Note the four approximately helical structures. Unlike the trajectory in Figure 1, there is no**
**obvious organization of the trajectory with time. (Light and dark points are distributed all over the attractor.)**



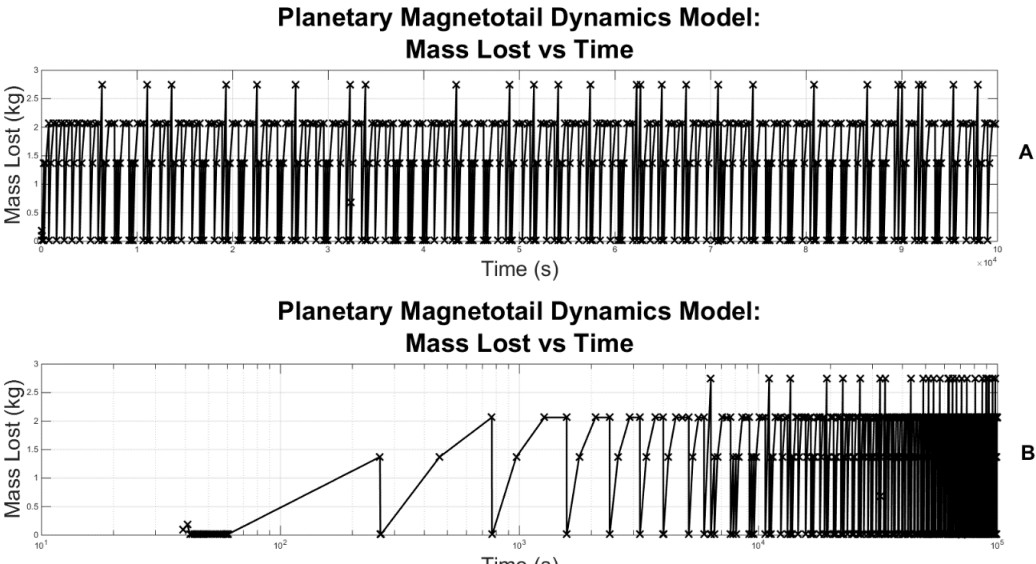

**Figure 5: Mass lost verses time plot for a strange attractor. A: linear time scale. B: Logarithmic time scale. The mass is lost at discrete times marked by crosses. The line linking the crosses is a visual aid. Note from panel A that the system is a-periodic (best seen from the irregular repeat times of the largest lost masses). Note from panel B that there is an initial transient period of between $10^2$ and $10^3$ seconds before the system arrives on the attractor.**


Three trajectories in phase space with small variations in initial conditions are shown in Fig. 6. The distorted helices are clearer in this line plot but time information is lost. The initial conditions for each run are shown in the legend of Fig. 6.

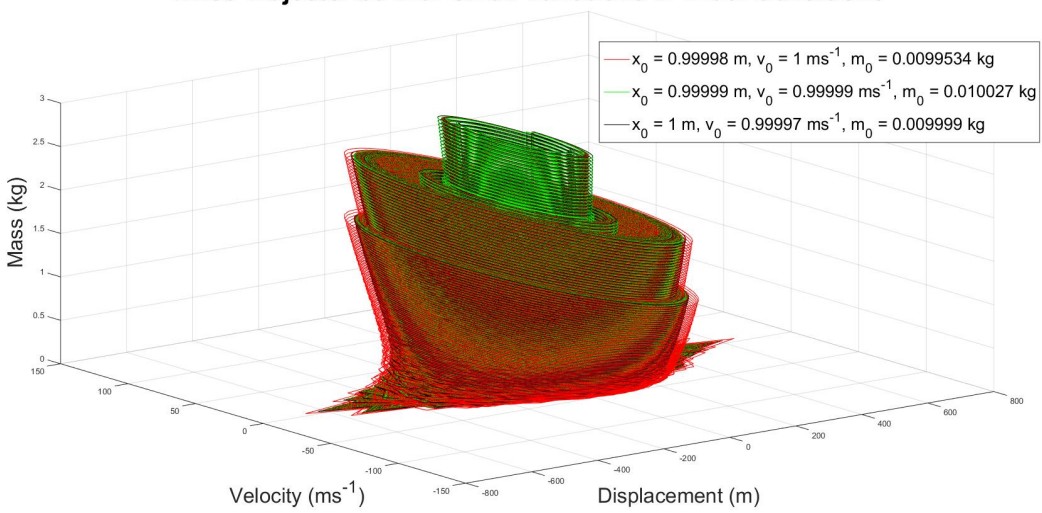

**Figure 6: Three trajectories in phase space. The legend shows the initial conditions for each of the three model runs: the variation in initial conditions is very small. The model parameters were constant across all three runs and are the same as shown in Table 3. All three runs arrive on the same attractor, which is the same as that seen in Figure 5.**

It can be seen that all three trajectories arrive on the same attractor but it is unclear how rapidly they diverge from each other across the attractor. How these separations evolve with time is illustrated in Fig. 7. All three trajectories remain very close in phase space until approximately 400 seconds into the run, when they diverge rapidly. Note that this is approximately the same time as it takes for the trajectory illustrated by Fig. 6 to arrive on the attractor. Since the attractor is the same (compare Figs. 4 and 6) and the variation in initial conditions is tiny, it can be assumed that the same length of time is taken for each of the trajectories to arrive on the attractor.

Once this transient period is over it takes only tens of seconds to arrive at $\Delta S > 100$ for the first time for all three separations of trajectories and less than 300 seconds for $\Delta S > 600$ to be exceeded for the first time by all three separations. At first glance it may appear from panel A of Fig. 7, that there is a periodic motion in the separations on a scale of 2000 – 3000 seconds. Closer inspection will show that in fact each cycle is not an exact repetition of any other. It can be seen (most easily in panel B of Fig. 7) that the separations undergo rapid oscillations of variable amplitude



but approaching $\Delta S = 0$ at each minimum, with periods of 10-20 seconds. This can be understood by considering the motion projected onto the velocity-displacement plane. Each trajectory will perform a series of approximate ellipses in this plane but with different "velocities" in phase

space. The trajectories will therefore "lap" each other in this plane and every time this occurs the separation in this plane reaches a minimum. The actual minimum separation in the full phase space will depend on the mass and on which of the attractor's four approximate helices the trajectories are on. The lowest separations will occur when two trajectories are on the same approximately helical structure in the attractor.

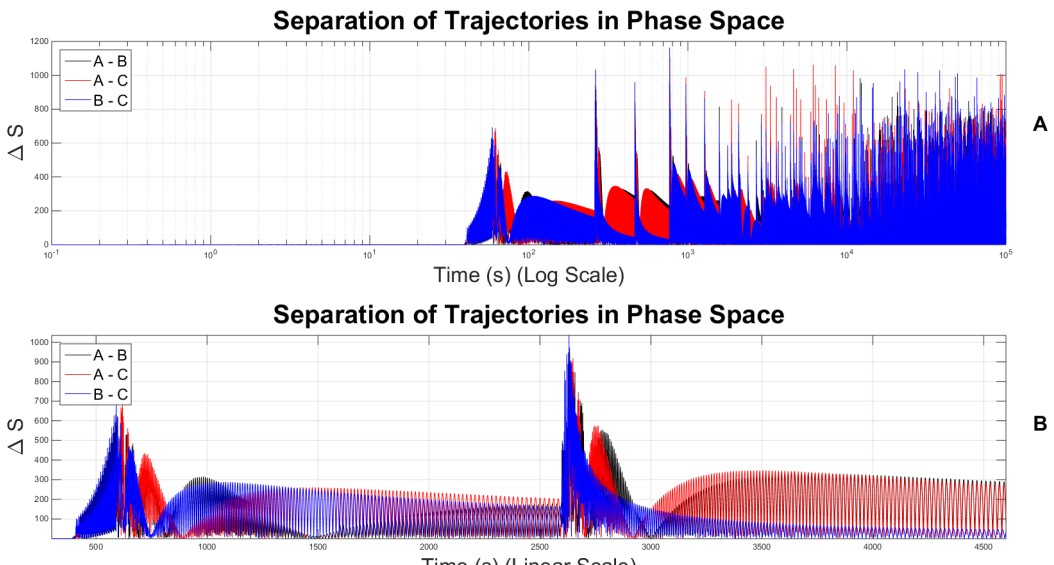

**Figure 7:** The evolution of the separation of trajectories, A, B and C with time. Panel A: Evolution over the entire model run on a log scale. Panel B:
Evolution between 300 and 4600 seconds on a linear scale. At approximately 400 seconds the trajectories suddenly diverge then fluctuate dramatically for the remainder of the run (panel A). What at first glance seems to be a periodic behavior on the scale of ~2000 to 3000 seconds can be seen, on closer inspection, never to repeat exactly. The fine structure of the evolution is more apparent in panel B, where individual short time-scale oscillations in the separations can be seen. See main text for an explanation of this behavior.



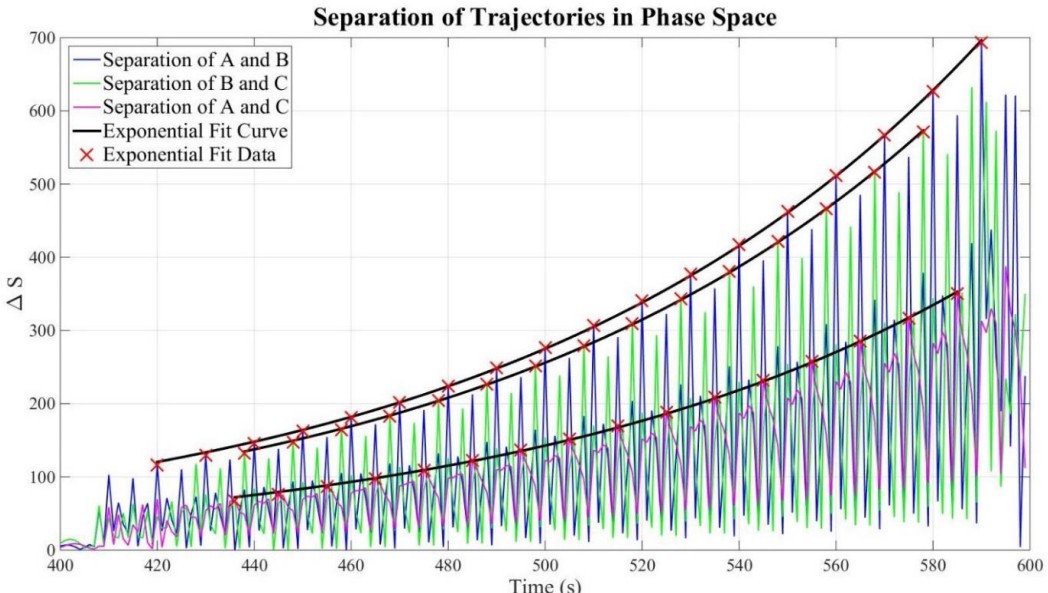

**Figure 8: The separations of three nearby trajectories (thin blue, green and magenta lines). These trajectories all arrive on the strange attractor at approximately 410- 430 seconds. The outer envelopes of the three separations, once on the attractor, are marked by red crosses. Exponential fits to the data marked by these crosses are shown by the thick black lines. The excellent fits of the exponential curves to the data is strongly indicative of deterministic chaos.**

The graph shown in Fig. 8 shows how the separations of the example trajectories behave during the crucial period immediately following arrival

on the attractor. The red crosses show points on the outer envelope of the separations which were used by an exponential curve fitting routine. The

fitted curves are shown by the thick black lines. Each is an excellent fit to the data. Thus, sensitive dependence on initial conditions is established,

for this set of example trajectories, which were randomly chosen.

**3.2 The effect of increasing the total loading rate, $\frac{dm}{dt}$, (Eq. ( 2 ) ).**

Changing the mass loading rate, which is analogous to changing loading by the Dungey and Vasyliunas Cycles, whilst keeping all other parameters

constant, may switch the system from limit cycle behaviour to chaotic behaviour. This is illustrated in Fig. 9 where the mass lost versus time is

plotted for two model runs that differ only in the total rate of mass increase, $C_D + C_V$. As for previous plots of this type, the discrete lost masses are





marked by crosses and the lines linking them are an aid to the eye. The results shown in black are for a model run with an order of magnitude larger combined rate of increase of mass than was used for the model run with the results shown in red. The former (black) case shows a-periodicity indicative of chaos, where-as the latter (red) case shows rapid settling to periodicity indicative of a limit cycle.

This type of behaviour, where increasing the value of a parameter that can be considered as a driving or controlling aspect of the system being modelled leads to switching from periodic to chaotic modes of behaviour, is very common. This is illustrated by the famous orbit diagram for the logistic map, $x_{n+1} = rx_n(1 - x_n)$, (Fig. 10).

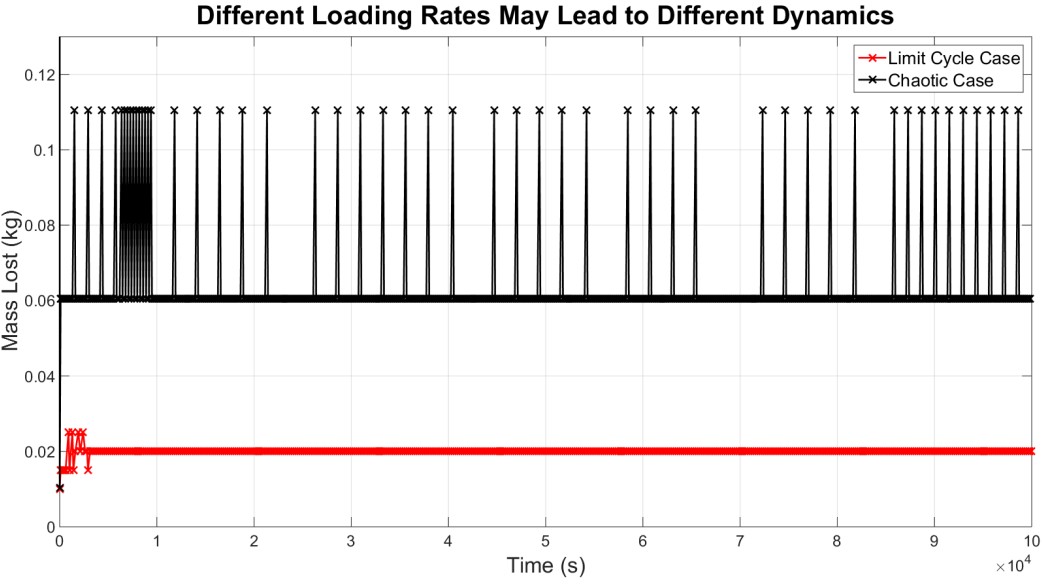

**Figure 9: Mass lost versus time in a limit cycle case (red) and chaotic case (black). The only difference between the two cases is that the total loading rate $(C_D + C_V)$ is altered. See Table 4 for the parameters and initial conditions used for these model runs.**



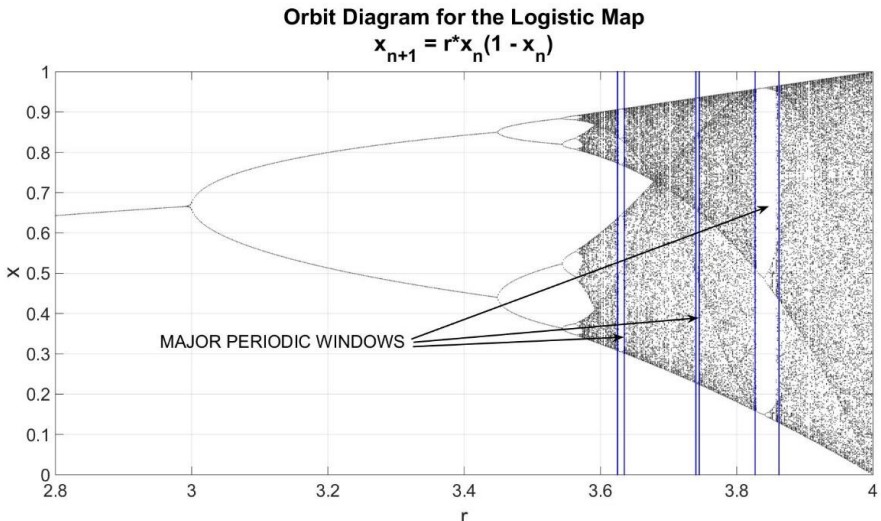

**Figure 10: The famous orbit diagram for the logistic map. Three periodic windows in the chaotic regime are marked by vertical solid lines.**

As the control parameter, $r$, is increased, the period doubles repeatedly until arrival at a chaotic state ($r \approx 3.57$). As $r$ is further increased, "periodic windows" appear, where chaos temporarily halts and periodic behaviour is restored before chaos resumes at an even higher value of $r$. The three most obvious of these are shown bracketed by vertical lines in Fig. 10. Qualitatively similar orbit diagrams can be computed for many systems. Flows (continuous time) such as the model presented here can be converted to maps using appropriate Poincaré sections. When this is done, similar orbit maps can be generated, showing the same type of chaotic regime with periodic windows. There is evidence (not shown) that the model

presented here can be driven repeatedly through such periodic windows by increasing $\frac{dm}{dt}$. The precise choice of $C_D + C_V$ is therefore not crucial for illustrating this behaviour. Nor is there anything particularly unique about the choice of the other model parameters; many other sets of values reproduce the same qualitative behaviours described above.

### 3.3 Comparison of the effects of dynamical noise and simulated instrument noise

   Figure 11 shows the results for the instrument noise simulations. The no noise case is represented by the black * symbols. The lower finite noise

case is represented by the red + symbols. The higher noise case is marked by the blue x symbols. The four bands with different values of $\Delta m$ seen





in the no noise case are broadened vertically by the addition of noise, exactly as would be expected, but at the levels of noise applied are still readily identifiable.

Figure 12 shows the equivalent results for the dynamical noise runs. The no noise (black *) case is repeated for reference and is identical to that
shown in Fig. 11. The lower but finite noise (red +) case shows a markedly different character to its equivalent in Fig. 11. Instead of an obvious broadening of the four bands seen in the no noise case, a new, fifth, band is observed. The timing of the creation of lost mass is also altered compared to the no noise case. The higher noise (blue x) case sees the appearance of a sixth band. The bands are now also noticeably broader than the no noise case. This increase in the complexity of the dynamics is also reflected in the trajectories through phase space of the noisy cases, as portrayed in Fig. 13, where panel A shows the no noise case, Panel B the low noise case and Panel C the high noise case, each corresponding to the examples
in Fig. 12. The number of approximately helical structures is seen to increase from Panel A to Panel B to Panel C just as the number of bands in Fig. 12 is seen to increase with increasing dynamical noise. The three cases of Fig. 13 are not recognisably on the same attractor. Comparing this with the three trajectories shown in Fig. 6, which differ only in the initial conditions and all arrive on a single strange attractor, it is seen that dynamical noise can have significant effects even when it is underlain by a fully deterministic system.

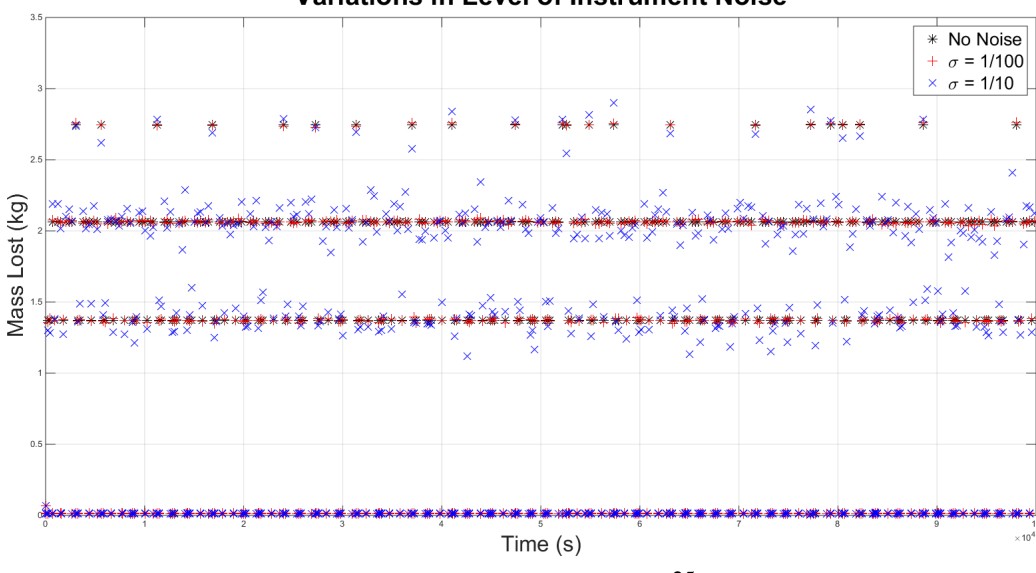





**Figure 11: The effects of simulated instrument noise. The four bands seen in the no noise case are broadened by the addition of noise. Compare this with Figure 12.**

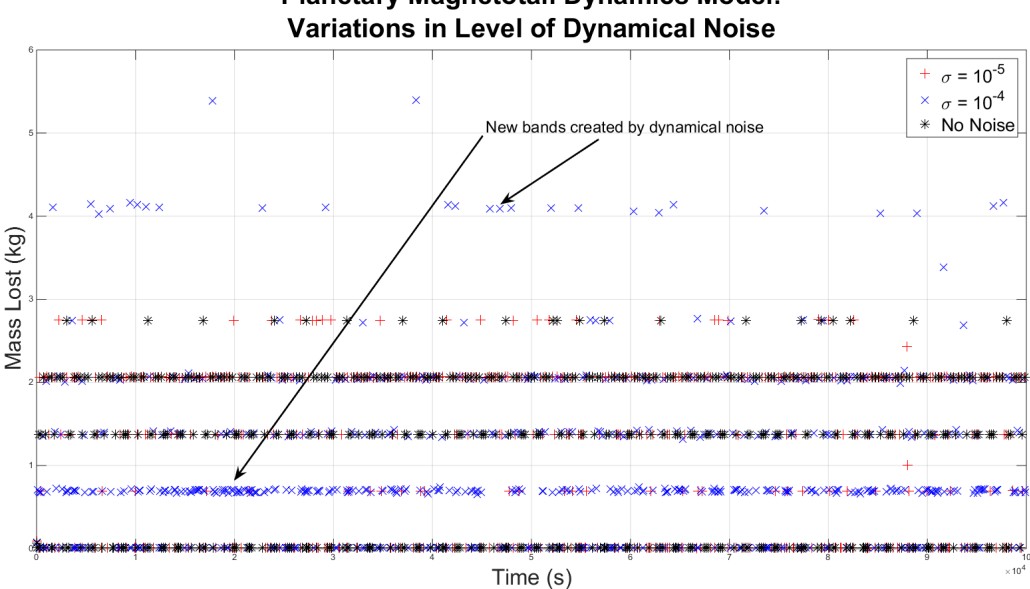

**Figure 12: Mass lost verses time for the same initial conditions and model parameters, but with different levels of dynamical noise. Note the increase in the number of discrete bands. Compare this with Figure 11 showing the effect of instrument noise.**


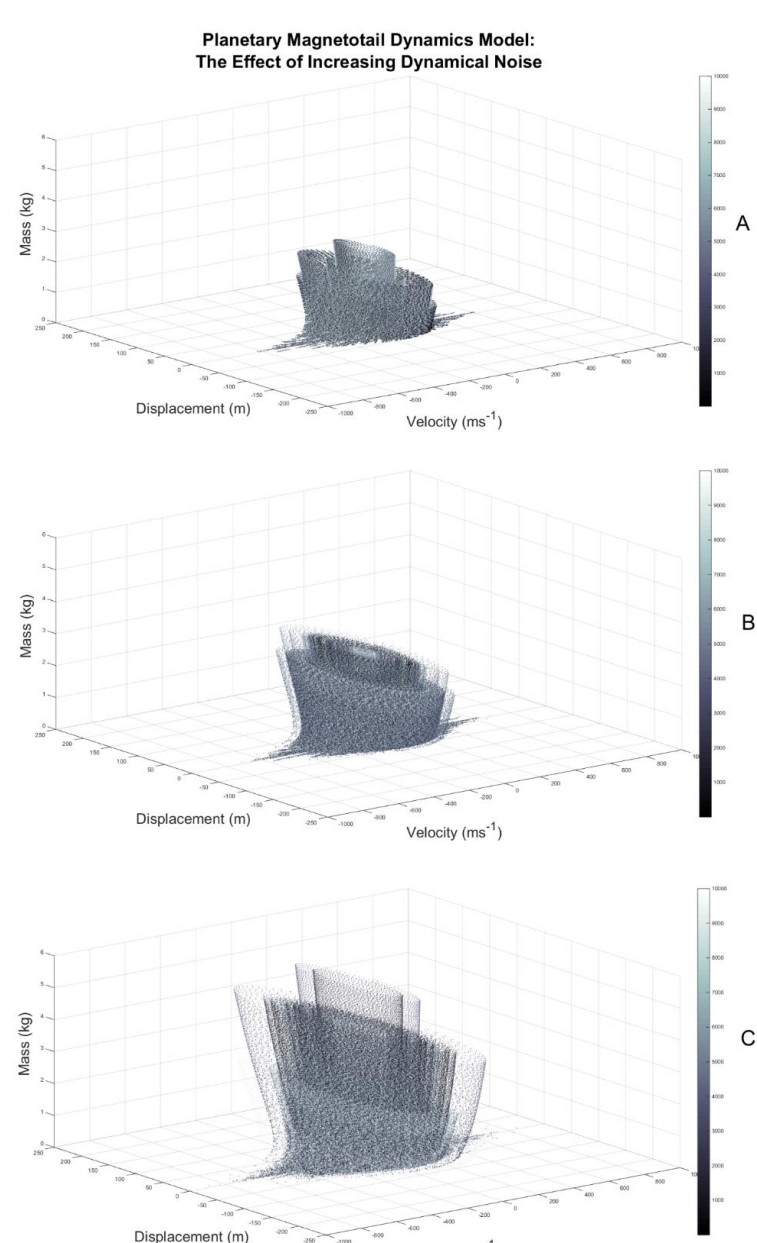




**Figure 13: Dynamical noise affects the geometry of an attractor in phase space. Panel A: the attractor without dynamical noise, corresponding to the black * case in Figure 12. Panel B: The low finite noise attractor corresponding to the red + case in Figure 12. Panel C: The higher noise case attractor corresponding to the blue x case in Figure 12. As the dynamical noise increases, so does the number of approximately helical structures in the attractor. Even the lower dynamical noise case is not recognisable as the same attractor as the no noise case.**

## 4 Conclusions


A model of global magnetotail dynamics in Earth-like cases was presented. The model is a relaxation oscillator representing a simple mechanical analogy to the magnetospheric case. It is an extension of the Shaw Model of leaky taps and as such it offers qualitative but not quantitative insight into the question of the types of dynamical behavior that Earth-like magnetotails might undergo. The model is capable of representing the Dungey and Vasyliunas Cycles, making it applicable to Mercury, Earth, Jupiter and Saturn, as well as a wide variety of hypothetical exoplanet cases. The

model is also capable of representing variable levels of dynamical noise within each Cycle. This is crucially important because the concepts of non-linear deterministic behavior and of stochastic behavior are not binary and mutually exclusive as has usually been tacitly assumed in this context. In fact, any level of dynamical noise may overlay a deterministic process, possibly masking the latter completely but equally possibly only modifying a recognizably deterministic evolution.

The model exhibits two main types of dynamical behavior:

A. Evolution towards an attracting limit cycle. (A closed loop in phase space that, once reached, is never escaped.) The dynamics, once a limit cycle is attained, is equivalent to periodic behavior.

B. Deterministic chaos after a finite transient period. Once the transient period is over, the motion in phase space never repeats exactly, but instead forever evolves within a confined region of the phase space known as a strange attractor. For the example shown, if the parameters of the model

are kept constant, then small (order of 0.1%) variations in the initial conditions lead to exponential divergence of trajectories in phase space. This is extremely strong evidence of deterministic chaos.

Switching between limit cycle and chaotic dynamics can be achieved by increasing the sum of the components on the right-hand side of Eq. ( 2) whilst leaving all other parameters the same. This is significant because the rate of increase of mass is analogous to the rate of build-up of stored

energy in the magnetotail case. It is therefore reasonable to conclude that the timing of plasmoid releases can vary from periodic to chaotic, as the rate of energy input increases.





The importance of considering a Noisy Chaotic Magnetosphere Hypothesis has been emphasized. The results given in section 3.3 show that sufficient *dynamical* noise has a different character to *measurement* noise in this system. Instead of steadily broadening a fixed number of bands of lost masses, as dynamical noise is increased, new bands are added. The most likely explanation for this phenomenon is that the model is being switched between different attractors by sufficiently different rates of mass increase. There is no easy way, given only a time series of a system variable, to detect the presence of dynamical noise. Instrument noise is much easier to detect and account for.

Even this simple, qualitative model has implications for the study of Earth-like magnetotails.

## 4.1 Earth

The model indicates that it is likely that as solar wind driving is increased, magnetotail behavior will go from a quiescent state to periodic plasmoid formation to chaotic plasmoid formation and possibly back to periodicity with a further return to chaos, as exemplified by the Logistic Map orbit diagram (Fig. 10). There is some observational evidence for this already (Baker et al., 1990). The model also suggests that input dynamical noise can have a fundamentally different effect on the system from output (observational) noise. The character of the dynamical noise effect shown here would be very difficult to disentangle from the underlying deterministic dynamics using only a series of observations of dynamical variables.

The chaotic regime poses challenges for forecasting. Because of the exponential divergence of arbitrarily close but not identical initial conditions, detailed forecasts are intrinsically restricted to the short term. Since chaos would be expected at the higher end of possible solar wind driving there is an unfortunate correlation between the least forecastable times and most desirable times to forecast. (Higher energy events have the greatest potential technological impacts.) Additionally, it can be expected that with fluctuations in solar wind driving of sufficient amplitude, the system can be switched between different modes of behavior – from one phase space attractor to another – leading to even greater short-term unpredictability. This dynamical noise makes effect forecasting even more difficult. Note that this forecasting problem does not relate to when plasmoid formation will start after a quiescent period but to behavior within an active period.

## 4.2 Planets

### 4.2.1 Mercury

At Mercury the Dungey Cycle dominates over the Vasyliunas Cycle and solar wind conditions vary significantly over time. This is qualitatively similar to Earth and it should be expected that plasmoid formation behavior will vary in its qualitative dynamics in similar fashion. Periodic and chaotic dynamics are likely to exist under different solar wind driving conditions. Dynamical noise at some scale is probable.



### 4.2.2 Jupiter

At Jupiter the Vasyliunas Cycle dominates over the Dungey Cycle. Since the angular velocity of the plasma can be assumed constant in this context, it should be expected that there is one dominant type of dynamics describing plasmoid formation. Whether that type is periodic or chaotic cannot be judged from a qualitative model such as the one presented here. Variations in solar wind driving would be expected to only cause small variations away from the dominant dynamics and may not be distinguishable if the dominant mode is chaotic. In this circumstance it might be possible to treat the Dungey Cycle contribution as dynamical noise variations to the Vasyliunas Cycle.

### 4.2.3 Saturn

Neither Cycle is negligible at Saturn. The total contribution from the Dungey and Vasyliunas Cycles will dictate the dynamics at any given time. It is likely that variations in solar wind driving cause variations in the plasmoid formation dynamics, as at Earth. Multiple periodic modes and chaotic behavior are all plausible. Dynamical noise in the Dungey Cycle is highly probable.

### 4.3 Exoplanets

Whilst the combinations of magnitudes of variables that dictate the relative and absolute strengths of the Dungey and Vasyliunas Cycles at any hypothetical exoplanet is practically infinite, this model suggests that the types of qualitative dynamics of plasmoid formation are not. The fundamental types are quiescent, periodic and chaotic. Additionally, variation in stellar wind driving may lead to switching between different periodicities and between periodic and chaotic behavior (dynamical noise).

**Author contribution**

Robert J. Burston conducted all work reported in this paper and wrote the paper itself.

**Competing interests:**

The author declares that he has no conflicts of interest.



**Acknowledgments**

The author gratefully acknowledges the support of the Science and Technology Facilities Council in supporting this work under grant
ST/P004016/1.

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
