# Peer review of ""Earth-like" planetary magnetotails as non-linear oscillators"

_Annales Geophysicae, 2020_

## Referee Comment (RC1) · Anonymous Referee #1 · 19 May 2020

Comment on <"Earth-like" planetary magnetotails as non-linear oscillators > by Robert J. Burston

This paper presents a non-linear oscillator model of a simple system analogous to "Earth-like" magnetotail plasmoid formation and release dynamics. I read this paper initially with great interest, but feel very disappointed in the end. Authors argued that the dynamics of "Earth-like" magnetotail can be studied by analogy with the movement of mechanical spring, but the model parameters for the run are set artificially (see table 1- table 6), which have nothing to do with the planetary magnetosphere. So, I cannot see any application to the planetary magnetotail, and cannot recommend publication. Below are my comments. Major comments: 1. The "Introduction" is too long. For example, the section 1.3.2, 1.3.3 texted here are nearly irrelevant to planetary magnetotail. Much of the content is about the fundamentals of dynamic system, like Chaos, limit cycles. I suggest deleting these irrelevant texts, and highlighting your reasons and motivations of this study. 2. The analogy is a good way to insight the dynamic physics of magnetotail. However, before performing the analogy, we have to show the physical reasons why the dynamics of planetary magnetotail can be studied by analogy with the spring. Currently, we actually know few about the true physics of planetary magnetotail, although many magnetotail substorm models have been presented. To what extent the analogy can well account for the magnetotail dynamics? The simple or casual analogy makes no sense to understand the magnetotail dynamics? 3. In this paper, author just performed the run for the dynamic behavior of a spring using artificial parameters, and argued the application to planetary magnetotail. Actually, I didn't see any parameters of planetary magnetotail are adopted to run the model. Therefore, this paper may fit the journals of nonlinear process better. 4. The author proposed that this model includes the Dungey and Vasliunas Cycle, and can separate the contribution from these two drivers in the Abstract ("It includes, for the first time in such a model, separate drivers for the Dungey and Vasyliunas Cycles. . ."). However, from Section 3.2, the results displayed here are only sensitive to the total contribution (CD+Cv), the author didn't show how can we separate these two drivers. Another question is that if we just change the value of CD+Cv, whether the magnetotail behavior shows similarly to the Logistic map orbit diagram (such as three periodic windows)? Is there any observational evidence? 5. Line 45: Note that Earth's magnetotail can also undergo Dungey cycle, and substorm can occur under northward IMF.

---

## Author Comment (AC1) · 24 May 2020

Response to the initial, general comment: "Authors argued that the dynamics of "Earth-like" magnetotail can be studied by analogy with the movement of mechanical spring, but the model parameters for the run are set artificially (see table 1- table 6), which have nothing to do with the planetary magnetosphere. So, I cannot see any application to the planetary magnetotail, and cannot recommend publication." The connection between the spring model and "Earth-like" magnetotails is discussed at length in section 1.3. I am not the first to make this analogy, as referenced in the first sentence of section 1.3.1. In lines 190-193, I clearly state that the model is qualitative, not quantitative: 'The new model. . .cannot offer quantitative results, the strong analogy with plasmoid formation and release allows for examination of the range of possible qualitative dynamics

that can be expected in Earth-like magnetotails.' Hence the reviewer is simply ignoring the entire argument set out in the Introduction, that the study of analogous systems is normal and fruitful in physics; has been used in the specific context of Earth's magnetotail before; that qualitative dynamics are a useful area of study. The reviewer does not make an actual argument supporting their point of view, but instead disguises the lack of such an argument by using the word, "so. . ." There is no real logical connection between the first and second sentences quoted above. There is no attempt to refute the argument that there is an appropriate analogy between leaky taps, non-linear springs and Earth-like magnetotails, an analogy that, I re-iterate has been used in previous publications by respected authors (Hones, Baker.)

Comment 1: Section 1.3.2 deals with previous work on analogies to magnetotail dynamics and therefore, far from being "irrelevant" is extremely pertinent to the matter at hand. Section 1.3.3. explains why studying analogous models that are simpler than the real system is a worthwhile activity and is, again, crucial, not "irrelevant." The section (1.4) on qualitative dynamics is included because the paper is about qualitative dynamics – a topic that, in my considerable experience of talking to space scientists about it, is not one that it is safe to assume is common knowledge at even an introductory level. Hence I explain the principles needed to understand the results of the paper, in the Introduction to the paper, so that readers have a reasonable chance of understanding my conclusions without resort to a text on dynamical systems theory. Again, essential, not "irrelevant." My motivation for the study is covered extensively already, in sections 1.1 through 1.3.

Comment 2: "The analogy is a good way to insight the dynamic physics of magnetotail." This contradicts what is said further down: "The simple or casual analogy makes no sense to understand the magnetotail dynamics?" Does the reviewer think the analogy is good or bad? No clear argument is presented either way. The physical reason for using the analogy is that the spring model presented is a relaxation oscillator and plasmoid release from Earth-like magnetotails qualitatively behaves like a relaxation

oscillator. This is explained in Section 1.3.1. The extent to which the model can account for Earth-like magnetotail dynamics is this: Qualitatively, only. But at that level, very well. See sections 3 and 4. Additionally, the model is intentionally "simple", see lines 69-74 and section 1.3.3. It is not, however, in any way "casual." Comment 3: The author did not "just" do anything. The reviewer again disguises the lack of a real argument, this time using "therefore," instead of, "so." The reviewer makes absolutely no attempt to explain why these qualitative results are invalidated simply because they are not quantitative. I make no claim that they are quantitative; I claim that, because of the analogy that the reviewer makes no attempt to refute, the behaviours seen in the model can be expected in Earth-like magnetotails. This paper is entirely inappropriate to journals specifically about non-linear processes because it innovates in regard to planetary magnetotails, not analysis of non-linear systems.

Comment 4: The reviewer confuses inputs and outputs: The model has separate drivers for the Dungey and Vasyliunas Cycles, as explained in section 2.1. These are inputs. I do not claim or even imply anywhere in the paper that the model has separate outputs for each cycle. This is because it does not. I do have evidence that that the model can be driven through the same kind of limit-cycle-to-chaos sequences as shown in the logistic map orbit diagram, however a clear demonstration of this would extend an already long paper by a very considerable amount so the necessary results are not shown, beyond the fact that simply increasing $C_d + C_v$ can switch from limit-cycle to chaotic behaviour. This is recognised already in lines 429-434.

Comment 5: Line 45 does not even mention the Dungey cycle, let alone suggest that it is not present in Earth's magnetosphere. My understanding is that substorms follow a flip to southward IMF. This can be short-lived compared to the timescale of a complete substorm and the IMF can have flipped back to Northward before the substorm is complete. Hence conditions of southward IMF are essential for substorms, but they do not have to last the entire duration of a substorm. I have modified lines 45-49 to reflect this.

---

## Referee Comment (RC2) · Anonymous Referee #1 · 26 May 2020

In your paper, you actually just used a set of artificial parameters to calculate the movement of a mechanical of spring based on Eqs. (1-3), and argue that your calculation can qualitatively represent the dynamics of magnetotail to separate drivers for the Dungey and Vasyliunas Cycles, and describe the detachment of plasmoid. Unfortunately, nothing about the planetary magnetotail is involved in you calculation, although you mentioned the possible applications in Earth and planets in subsection 4.1 and 4.2. If I were you, I would input the typical parameters of planetary magnetotail into your model(are equivelent spring parameters easy derived?) to run the dynamics of spring analogue, and compare the output with some space simulations to illustrate the validity and reasonability of the spring model. Anyway, the current manuscript is too premature to be published.

---

## Author Comment (AC2) · 26 May 2020

The reviewer replaces argument with pejorative language, which is entirely unacceptable: I re-iterate that I did not "just" do anything. Nothing is "unfortunate" about this paper and there absolutely is no sense in which it is "premature." These terms belittle my work without providing any logical or scientific argument refuting anything in the paper. If one removes the pejorative term "just" the first sentence of the reviewer's latest comment is simply a statement of fact. There is no argument about anything in it. Similarly, in the second sentence, if the pejorative word, "Unfortunately," is removed, it becomes a statement of fact. In the final sentence, if the word "premature," is removed it becomes completely meaningless. None of these represents an argument. The remainder, beginning, "If I were you," again completely ignores, rather than attempts to

refute, the argument and evidence presented that studying analogous systems is a useful contribution to science. It also insists, without justification, that there should be some kind of one-to-one mapping between the spring model and the magnetotail for it to be useful. If such a thing existed there would be no point to the spring model, as one would have a model of the magnetotail. As for comparison with models of the magnetotail, that is insisting that this theory paper become an experimental paper without justifying why I should do so: Theory papers and experimental/observational papers have been considered acceptable as separate entities in physics for centuries. Since the analogy has been used before in articles by respected authors in respected journals, I have no real need to justify the use of the analogy now, but I have done so at length, already. In conclusion, the reviewer has twice failed to provide any attempt at a logical or scientific argument refuting the content of the paper, nor produced any justification at all for recommending rejection of the paper but has instead resorted to a subtle form of abuse i.e. pejorative language.

---

## Referee Comment (RC3) · Massimo Materassi (Referee) · 10 Jun 2020

The paper by Dr Burston is definitely interesting and timely, as finally the community is considering complex dynamics and stochastic forcing as important elements of space plasma physics. I personally approve the manuscript as it is, being it clear and interesting, being the description of what Dr Burston claims to be doing well written and reader-stimulating. I just suggest Dr Burston to have a look at Chapter 14 of the recently published book "The Dynamical Ionosphere", by Elsevier, that I have worked for as an Editor: in that Chapter I introduce noise terms to describe ionospheric turublence, and suggest "regular" path integral techniques to treat the consequent dynamics. For sure, applying the same things to the model of Dr Burston would be interesting in future research. In that Chapter 14, it is also considered how to alter the dynamics of

a parcel of fluid in the presence of mass variability with time (over there, this is due to chemical reactions, as recombination or ionization creating different chemicals): one might suggest to alter equation (1) of Dr Burston's paper accordingly.

Best regards

Massimo Materassi, PhD CNR-ISC
* * *

---

## Author Comment (AC3) · 11 Jun 2020

I appreciate the positive comments and will seek out the book chapter referred to.